# Efficacy of Remdesivir and Neutralizing Monoclonal Antibodies in Monotherapy or Combination Therapy in Reducing the Risk of Disease Progression in Elderly or Immunocompromised Hosts Hospitalized for COVID-19: A Single Center Retrospective Study

**DOI:** 10.3390/v15051199

**Published:** 2023-05-19

**Authors:** Davide Fiore Bavaro, Lucia Diella, Alessandra Belati, Giuliana Metrangolo, Laura De Santis, Vito Spada, Michele Camporeale, Angelo Dargenio, Gaetano Brindicci, Flavia Balena, Deborah Fiordelisi, Fabio Signorile, Giacomo Loseto, Crescenza Pasciolla, Carla Minoia, Immacolata Attolico, Tommasina Perrone, Simona Simone, Maria Rendina, Nicoletta Giovine, Francesco Di Gennaro, Pellegrino Musto, Attilio Guarini, Alfredo Di Leo, Loreto Gesualdo, Maria Dell’Aera, Annalisa Saracino

**Affiliations:** 1Clinic of Infectious Diseases, Department of Precision and Regenerative Medicine and Ionian Area, University of Bari, Piazza G. Cesare 11, 70124 Bari, Italy; 2Hematology Unit, IRCCS Istituto Tumori “Giovanni Paolo II”, 70124 Bari, Italy; 3Unit of Hematology and Stem Cell Transplantation, AOUC Policlinic, 70124 Bari, Italy; 4Nephrology Dialysis and Transplantation Unit, Department of Precision and Regenerative Medicine and Ionian Area, University of Bari, 70124 Bari, Italy; 5Section of Gastroenterology, Department of Emergency and Organ Transplantation, University of Bari, 70124 Bari, Italy; 6Hospital Pharmacy Department, University Hospital of Bari, 70124 Bari, Italy; 7Department of Precision and Regenerative Medicine and Ionian Area University of Bari and Unit of Hematology and Stem Cell Transplantation, AOUC Policlinico, 70124 Bari, Italy

**Keywords:** SARS-CoV-2, remdesivir, sotrovimab, antivirals, monoclonal antibodies, severe COVID-19, immunocompromised hosts, combination therapy, elderly, COVID-19 progression, COVID-19 therapy

## Abstract

Introduction: Remdesivir (REM) and monoclonal antibodies (mAbs) could alleviate severe COVID-19 in at-risk outpatients. However, data on their use in hospitalized patients, particularly in elderly or immunocompromised hosts, are lacking. Methods: All consecutive patients hospitalized with COVID-19 at our unit from 1 July 2021 to 15 March 2022 were retrospectively enrolled. The primary outcome was the progression to severe COVID-19 (P/F < 200). Descriptive statistics, a Cox univariate–multivariate model, and an inverse probability treatment-weighted (IPTW) analysis were performed. Results: Overall, 331 subjects were included; their median (q1–q3) age was 71 (51–80) years, and they were males in 52% of the cases. Of them, 78 (23%) developed severe COVID-19. All-cause in-hospital mortality was 14%; it was higher in those with disease progression (36% vs. 7%, *p* < 0.001). REM and mAbs resulted in a 7% (95%CI = 3–11%) and 14% (95%CI = 3–25%) reduction in the risk of severe COVID-19, respectively, after adjusting the analysis with the IPTW. In addition, by evaluating only immunocompromised hosts, the combination of REM and mAbs was associated with a significantly lower incidence of severe COVID-19 (aHR = 0.06, 95%CI = 0.02–0.77) when compared with monotherapy. Conclusions: REM and mAbs may reduce the risk of COVID-19 progression in hospitalized patients. Importantly, in immunocompromised hosts, the combination of mAbs and REM may be beneficial.

## 1. Introduction

Since the early days of the COVID-19 pandemic, the pathogenesis, clinical manifestations, and possible treatment strategies for this novel viral infection have been deeply investigated.

Initial research was focused on multiple drugs, including corticosteroids [1], immunomodulatory therapy [2], Chinese herbal medicine [3], natural products [4], and monoclonal antibodies [5,6,7], which were targeted against the most clinically relevant phase of the disease, the so-called hyperinflammatory phase corresponding to the “cytokine storm”.

Subsequently, after the first pandemic wave and the documented ineffectiveness of older drugs (including lopinavir/ritonavir, hydroxychloroquine, and ivermectin) [8,9,10], interest moved to monoclonal antibodies and drugs with direct antiviral activity that were potentially able to stop viral replication and consequently reduce the risk of progression of COVID-19 from the viral phase to severe lung failure. Currently, the monoclonal antibodies for mild to moderate non-hospitalized patients at high risk of progression are represented by sotrovimab, bamlanivimab/etesevimab, casirivimab/imdevimab, bebtelovimab (not available in Italy), and tixagevimab/cilgavimab [11,12]; on the other side, antiviral drugs are represented by remdesivir (for intravenous use), molnupiravir, and nirmatrelvir/ritonavir. To date, all these drugs have proved to be effective in reducing the risk of severe disease in patients with early COVID-19 [13,14,15,16]. Despite this growing armamentarium, evidence regarding the role of these treatments in the management of hospitalized patients with early COVID-19 still needs to be explored, especially in a “real-life” setting, including patients underrepresented in clinical trials (e.g., immunocompromised hosts). Indeed, current European guidelines for the management of hospitalized patients limit the recommendation to casirivimab/imdevimab or remdesivir in subjects with lung failure [17]; by contrast, the National Institutes of Health (NIH) guidelines [18] have made a conditional recommendation on the potential role of remdesivir and short-course remdesivir (sc-REM) also in the hospital setting, with the same three-day administration schedule that is used for outpatients [15,19]. Still, the document does not produce any recommendation regarding the use of monoclonal antibodies for hospitalized patients or any specific suggestion for immunocompromised hosts.

Data regarding the efficacy and safety of REM in combination with monoclonal antibodies in hospitalized settings are limited to case reports [20]; additionally, the best treatment strategy for immunocompromised hosts with COVID-19 is a matter of debate [21].

Finally, the progressive de novo or in vivo emergence of viral variants showing resistance to antiviral therapies also needs to be addressed in the future [22], along with the development of new drugs for combating COVID-19 [23]. In this sense, it could be speculated that combination therapy could address drug resistance conferred by emerging variants, but data are still very preliminary [24].

Accordingly, the aim of this study was to evaluate the efficacy of monoclonal antibodies and remdesivir used as a mono- or combination therapy in reducing the risk of disease progression in hospitalized patients with mild to moderate COVID-19 in a “real-life” setting, involving old and very old patients with several comorbidities and including immunocompromised hosts.

## 2. Materials and Methods

### 2.1. Study Design

#### Patient Population

This is a retrospective cohort study including all consecutive patients who were hospitalized due to COVID-19 at our infectious disease unit from 1 July 2021 to 15 March 2022. The study was conducted in a 1200-bed tertiary care hospital in southern Italy (the Policlinico of Bari). In particular, our COVID-19 unit consisted of 28 beds dedicated to low-intensity and sub-intensive care.

The demographic characteristics of the patients, all their comorbidities, their COVID-19 signs and symptoms, their clinical presentation and laboratory findings on admission, their need for supplementary oxygen therapy including noninvasive or invasive ventilatory support, their secondary complications during hospitalization, and their outcomes (discharge or death) were retrieved from our internal database.

### 2.2. Treatment Exposure

In our hospital, a panel of experts elaborated an algorithm (Table 1) to guide the management of patients with COVID-19 that was regularly updated according to any new release from the scientific literature.

Along with standard care (including corticosteroids and low molecular weight heparin) from 1 November 2021, the use of mAbs (casirivimab/imdevimab until 15 December and sotrovimab thereafter) and sc-REM (from 12 January 2022) were included in our internal protocol for hospitalized patients. Particularly, sc-REM was prescribed to all subjects not in need of oxygen therapy and without the presence of risk factors for severe COVID-19, while mAbs were considered in cases of ineffective vaccination or when antiviral therapy was contraindicated or unavailable for any reason.

In the cases of patients presenting with radiological signs of pneumonia and in need of supplementary oxygen therapy, a standard 5-day course of REM was prescribed.

In the cases of an immunocompromised state or multiple risk factors in subjects with ineffective vaccination, the combination therapy (mAbs plus sc-REM) was also taken into consideration; in these cases, the two drugs were administered simultaneously.

Exclusion criteria from antiviral or mAbs therapy were symptom onset more than 10 days ago, requirement of HFNC, invasive or non-invasive mechanical ventilatory support, vasoactive drugs, or extracorporeal membrane oxygenation (ECMO). Contraindications to sc-REM/REM included aspartate aminotransferase and alanine aminotransferase ≥5 times the normal range values and a glomerular filtration rate <30 mL/min.

Notably, during the period of the study, oral antiviral therapies (nirmatrelvir/ritonavir and molnupiravir) were unavailable for use in the hospital setting; therefore, they were not included in the protocol.

### 2.3. Study Outcomes

The primary objective of the study was to analyze the efficacy of REM, sc-REM, and mAbs (exposure variables) in preventing progression to severe COVID-19 pneumonia. 

Disease progression was defined as the development of severe respiratory failure (P/F < 200) and a need for high flow oxygen therapy or noninvasive/invasive mechanical ventilation.

Secondary endpoints were severe adverse events to antiviral therapy or mAbs (requiring therapy discontinuation or medical treatment) and all-cause in-hospital mortality.

In order to include only episodes of severe respiratory failure related to COVID-19, the follow-up time for the primary outcome was set up to day 30 after the onset of symptoms.

### 2.4. Other Definitions

COVID-19 severity was defined according to (NIH) guidelines [18] as follows:
Asymptomatic or presymptomatic infection: individuals who test positive for SARS-CoV-2 using a virologic test (i.e., a nucleic acid amplification test [NAAT] or an antigen test) but have no symptoms consistent with COVID-19;Mild illness: individuals who have any of the various signs and symptoms of COVID-19 (e.g., fever, cough, sore throat, malaise, headache, muscle pain, nausea, vomiting, diarrhea, or loss of taste and smell) but do not have shortness of breath, dyspnea, or abnormal chest imaging;Moderate illness: individuals who show evidence of lower respiratory disease during clinical assessment or imaging and who have an oxygen saturation measured by pulse oximetry (SpO_2_) ≥ 94% in room air at sea level;Severe illness: individuals who have SpO_2_ < 94% in room air at sea level, a ratio of arterial partial pressure of oxygen to fraction of inspired oxygen (PaO_2_/FiO_2_) < 300 mm Hg, a respiratory rate >30 breaths/min, or lung infiltrates >50%;Critical illness: individuals who have respiratory failure, septic shock, and/or multiple organ dysfunction.

Patients were defined as “immunocompromised” if at specific risk for opportunistic infection: recent (within 1 year) solid organ transplantation, stem cell transplantation, prolonged neutropenia with absolute neutrophil count <500 cells/mL, CD4 cell count <200 cells/mL in HIV patients, or chronic corticosteroid and/or another immunomodulator therapy (e.g., anti-CD20 monoclonal antibodies) causing them to be at risk of severe infection. In addition, patients with solid cancer or hematologic neoplasia in active cytotoxic chemotherapy or subjects with congenital immunological disorders were considered “immunocompromised”.

Active cytotoxic chemotherapy was defined as any intravenous chemotherapy administered in the last 30 days before the SARS-CoV2 infection or any oral anti-neoplastic drug in course at the time of COVID-19 onset.

Patients were defined as having “ineffective vaccination” if they had received fewer than 3 doses of vaccination (including those who refused vaccination) or did not produce detectable IgG anti-spike against SARS-CoV2 due to an immunocompromised state or another immunological defect.

### 2.5. Statistical Analysis

Descriptive statistics were produced for the demographic and clinical characteristics of the patients, using the median and interquartile range (q1–q3) for non-normally distributed variables, and numbers and percentages for categorical variables.

According to the study outcome, a comparison between patients who developed a COVID-19 disease progression and those who did not was performed. The distribution of demographic, laboratory, and clinical characteristics between the two groups was analyzed by means of univariate parametric or nonparametric tests, using the Kruskal–Wallis or Mann–Whitney U tests (where appropriate) for continuous variables and the Pearson’s χ^2^ test (Fisher’s exact test where appropriate) for categorical variables, according to data distribution.

In order to assess the efficacy of different treatments and predictors of COVID-19 disease progression, a univariate Cox regression analysis was performed. A multivariate Cox analysis was made using a forward stepwise procedure, entering all variables with univariate *p* < 0.05 and those deemed clinically significant. Consequently, hazard ratios (HRs) (95% CIs) and adjusted hazard ratios (aHRs) (95%CIs) were calculated. In addition, a subgroup analysis was also performed to evaluate differences in the primary end point in patients with an immunocompromised state or ineffective vaccination.

Finally, Kaplan–Meier curves were built to compare time to disease progression for variables of interest. Nonparametric (log-rank) tests were used to compare event-free survival functions in the study groups.

Finally, an inverse probability treatment-weighted (IPTW) analysis was applied to estimate the average treatment effect (ATE) of REM or mAbs in reducing the risk of COVID-19 progression; it was calculated using the treatment effects module implemented in Stata 16.1 Special Edition.

Accordingly, the potential outcome for each patient was estimated using an average of the outcomes of similar patients who did not receive REM or mAbs.

To allow unbiased comparisons between groups, the final model was weighted as follows, according to baseline covariates influencing each patient’s risk of COVID-19 progression and treatment assignment, determined by means of the Cox multivariate model [20].

Stabilized weights (Appendix A) based on the inverse of the propensity score were applied to generate a weighted cohort in which the covariate distributions were independent of the treatment assignment [24].

Standardized differences were examined to assess balance using a threshold of 10% to indicate a clinically meaningful imbalance requiring further adjustment in outcome analyses [24]. The weighted propensity score distributions were visually inspected, and an overidentification test for covariate balance was performed to ensure that the final model and all covariates were balanced between groups, to allow valid comparisons. Similarly, the treatment group was conditioned on covariates that were not optimally balanced after IPTW (standardized difference ≥10%).

Finally, the IPTW effect regression was computed by taking the average of the difference between the observed and potential outcomes for each patient.

To further explore the potential for confusion due to differences in baseline infection complexity or patient health status, we conducted a number of prespecified subgroup analyses (age, presence of comorbidities, vaccination status, and immunocompromised state).

In all cases, a *p* value < 0.05 was considered statistically significant. The statistical analysis was performed using STATA “Special Edition” version 16.1 (STATA Corp., Lakeway Drive, TX, USA).

## 3. Results

### 3.1. Study Population and Outcome Distribution

Overall, 374 patients were hospitalized for COVID-19 during the study period. Of these, 43 were excluded from the treatment protocol with antivirals or mAbs due to critical illness at hospitalization requiring ventilation support.

Hence, a total of 331 patients were included in the final analysis; the median (q1–q3) age was 71 (51–80) years; 173 (52%) were males. Overall, 226 (68%) were fully vaccinated for COVID-19, and 25 (8%) received incomplete vaccination courses (fewer than three doses), while 80 (24%) did not receive any dose of anti-SARS-CoV2 vaccine.

At the time of admission to our unit, 91 (27%) were asymptomatic for COVID-19, and 113 (34%) presented with fever and constitutional symptoms, while 30 (9%) showed signs of pneumonia with a need for supplementary oxygen therapy.

According to the internal algorithm (Table 1), 185 (56%) patients were treated with corticosteroids for the management of lung failure. Finally, 88 (27%) patients received sc-REM, and 32 (10%) received the 5-day course of REM, while mAbs were prescribed in 37 (11%) cases, of which 18 (5%) received combination therapy with REM. 

In Figure 1 the retrospective enrollment flowchart is depicted.

During the study period, 78 (24%) patients developed disease progression (Table 2). These patients were more frequently older (median age 75 versus 66 years old, respectively, *p* < 0.005), unvaccinated, or not fully vaccinated against COVID-19 (79% versus 33%, *p* < 0.001) and presenting at admission with pneumonia and hypoxia (23% versus 5%, *p* < 0.001).

Interestingly, at the univariate analysis, no statistically significant difference in terms of the incidence of severe COVID-19 was evidenced in patients who received mAbs (5% versus 13%, *p* = 0.052). Conversely, an evaluation of the association between disease progression and antiviral therapy showed that a reduced number of patients exposed to sc-REM developed severe COVID-19 lung failure [11 (13%) versus 77 (87%) patients, post-hoc *p* = 0.004].

Importantly, no serious adverse event to REM or mAbs was recorded.

Finally, the crude all-cause in-hospital mortality was 14% and was significantly higher in those with disease progression (36% vs. 7%, *p* < 0.001) when compared with remaining subjects.

### 3.2. Predictors of COVID-19 Disease Progression in the Overall Population

In order to assess the independent predictors of COVID-19 disease progression, a Cox regression analysis was performed (Table 3).

According to our model, older age (aHR = 1.04, 95%CI = 1.01–1.06), hematological malignancies (aHR = 3.62, 95%CI = 1.00–13.05), mild–moderate (aHR = 4.70, 95%CI = 1.86–11.83) or severe COVID-19 at admission (aHR = 3.35, 95%CI = 1.11–10.12) and secondary severe infections in course of hospitalization (aHR = 3.36, 95%CI = 1.30–8.63) were independent predictors of COVID-19-related lung failure. By contrast, three doses of vaccination (aHR = 0.11, 95%CI = 0.05–0.24, *p* < 0.001) and sc-REM (aHR = 0.20, 95%CI = 0.08–0.49) were protective. Finally, the use of mAb (*p* = 0.104) and combination therapy (*p* = 0.236) had no significant impact on the outcome.

Consequently, Kaplan–Meier survival curves were made for different variables of interest. Notably, in this analysis, the use of sc-REM (log rank *p* = 0.027, Figure 2a) was associated with a reduced risk of progression, while ineffective vaccination at the baseline was associated with severe COVID-19 (log rank *p* < 0.001, Figure 2b).

The Individual impact of REM and mAbs on the risk of COVID-19 progression was analyzed by performing an IPTW regression analysis. The full cohort was balanced according to age, sex, incidence of secondary infections, disease severity at admission, and time from symptom onset to therapy (Appendix A). 

In the final model, the use of REM and mAbs was, respectively, associated with a 7% (95%CI = 3–11%) and a 14% (95%CI = 3–25%) reduction in the risk of severe COVID-19.

In addition, the subgroup analysis (REM: Figure 3; mAb: Figure 4) confirmed that the highest beneficial effect was evidenced in the case of older patients (>65 years old) with at least one metabolic comorbidity, ineffective vaccination, or an immunocompromised state.

### 3.3. Impact of Mono- or Combo-Therapy on Risk of COVID-19 Progression According to Immunological Status

A sub-analysis was made to explore the incidence of COVID-19 progression in the subgroup of patients with an immunocompromised state or ineffective vaccination. Interestingly, as shown in Figure 5, REM or mAbs use was associated with a significantly lower incidence of severe COVID-19 when compared to no therapy (38% vs. 59%, *p* = 0.009) and with additional benefit if used in combination (11% vs. 59%, *p* = 0.007); by contrast, these results were not confirmed in the cohort of immunocompetent hosts with effective vaccination.

A Cox regression model for predictors of COVID-19 disease progression was performed on this population. A total of 114 subjects were included in this analysis; 54 (47%) of them developed severe COVID-19. Interestingly, at univariate analysis (Table 4), both mAbs (HR = 0.19, 95% CI = 0.04–0.80) and REM (HR = 0.40, 95% CI = 0.14–0.70) in monotherapy were protective against the risk of progression of the disease. A multivariable analysis was performed, and in this case also, the results showed that mAbs or REM in monotherapy (aHR = 0.23, 95% CI = 0.08–0.65) were protective, but the combination therapy was the strategy associated with the lowest probability of failure (aHR = 0.06, 95% CI = 0.02–0.77).

## 4. Discussion

To the best of our knowledge, this is the first study that explored the efficacy of REM and mAbs in mono- and combo-therapy in a “real-life” cohort of hospitalized COVID-19 patients, including older adults and immunocompromised hosts. Interestingly, this study also highlighted the potential subgroup of patients who benefit from different therapeutic options, including sc-REM, REM, and mAbs, and mono- and combination therapies.

During the study period, the patients hospitalized and treated belonged to four main subgroups, including (i) immunocompetent and effectively vaccinated patients; (ii) older and frailer patients with multiple metabolic comorbidities; (iii) immunocompromised hosts; (iv) and subjects with ineffective vaccination. Given the observational nature of the study, the heterogeneity of the population in terms of the severity of COVID-19 at the baseline, and the prescription of therapies based on clinical judgment, an IPTW regression analysis was performed in order to balance the study cohort and evaluate more correctly the effect of the therapies.

According to the result of this study, the impact of antiviral or monoclonal therapies on the risk of progression of COVID-19 was stronger in patients belonging to the last three categories, while treatment was not significantly associated with a better outcome in immunocompetent and effectively vaccinated patients. Regarding the secondary outcome of this work, no serious adverse events to therapies leading to drug discontinuation or medical interventions were recorded. This is another important point for discussion: due to the high level of safety of antiviral therapies and mAbs for COVID-19, clinicians should be encouraged to treat patients at high risk. 

In discussing in depth the main outcome of the study, attention should be focused on the category of old patients, generally defined as adults who are over 65 years of age. This subgroup of patients is at a progressively increasing risk of severe COVID-19 for each additional year of age when compared with younger subjects [25]. This risk derives from the natural pro-inflammatory evolution of the immune system, known as “inflammaging” [26], that facilitates the occurrence of “cytokines storm”, severe lung failure, and/or thrombotic events in older patients [26]. In addition, it should be remembered that older age is usually associated with an increase in “frailty”, which is another important predictor of atypical COVID-19 presentation and worse outcomes [27] due to the low physiological chances of recovery from infection-related organ injuries. Consequently, it is not surprising that this category of patients gained a significant benefit from the use of antivirals or mAbs able to hamper viral replication and, in turn, the development of symptomatic COVID-19.

Moreover, based on our data, the coadministration of monoclonal antibodies and antiviral drugs could be an effective treatment strategy for the specific category of immunocompromised hosts. Indeed, it is known that these patients, particularly those affected by hematological diseases treated with anti-CD20 antibodies, present a high risk of persistent COVID-19, severe disease, and prolonged viral shedding, due to B-cell depletion associated with a significant reduction in cell-mediated immunity [28].

Immunocompromised hosts are usually excluded from clinical trials and registration studies; therefore, data regarding the efficacy and safety of several drugs in these patients are uncertain, as well as the optimal treatment and management strategies. In this regard, this work offers some interesting insights that could be the basis for improving clinical care and stimulating further studies.

In this specific setting, the adoption of different therapeutic strategies, including combination therapies (as in our experience) or prolonged treatments beyond the standard duration as described by some authors [29,30], may be useful, aiming to inhibit viral replication and enhance SARS-CoV-2 clearance by compensating for the impairment of natural cellular and humoral response. Conversely, a monotherapy with an antiviral or a monoclonal antibody could be adequate to reduce the risk of progression in immunocompetent and vaccinated patients. Nevertheless, the effectiveness of neutralizing monoclonal antibodies is influenced by the viral variant of concern (VoC), and the choice of therapeutic molecule should be based on these data, while antiviral drugs maintain their efficacy independently of the VoC [31,32]. Consequently, in this setting, antiviral therapies, such as remdesivir, may be the first choice in immunocompetent patients. Notably, this work confirmed the potential role of REM as a valid therapeutic option in cases of mild–moderate COVID-19 infection in hospitalized patients, particularly if administered during the first five days after the onset of symptoms [33,34,35]. However, few data are currently available on the effectiveness of short-course remdesivir for hospitalized patients not requiring oxygen therapy due to COVID-19. A recent work reports a case of a series of frail patients hosted in a non-acute care setting treated with sc-REM for healthcare-associated COVID-19 with encouraging results [36]. Similarly, even patients hospitalized for SARS-CoV-2 infection, with mild–moderate disease, may benefit from the same short-course treatment, as shown by the results of this study. Of note, in an exploration of the efficacy of REM in immunocompromised patients, both sc-REM and REM exerted the same efficacy, while in immunocompetent individuals the use of sc-REM was significantly more effective than REM when used after the onset of lung failure. This result could probably be explained by the fact that in immunocompromised hosts the viral replication could be persistent for many days even during the “hyperinflammatory phase” [28]; accordingly, in this setting, antiviral therapy may be beneficial even after the first days following the onset of disease.

On the other hand, the results of this real-life study enhance the data on the efficacy of monoclonal antibodies, with a particular emphasis on immunocompromised subjects or individuals with ineffective vaccination. However, this is unsurprising, since different clinical trials have been previously conducted on the role of monoclonal antibodies in hospitalized patients, demonstrating a greater efficacy in seronegative subjects [37,38,39]. Future studies should also address the potential role of other molecules in combination with current therapies, for instance, those deriving from nanotechnologies [40].

Excluding age, the second most important factor associated with the risk of lung failure was the severity of the clinical picture at presentation: as expected, mild–moderate and severe COVID-19 posed a significant risk of disease progression when compared with the pre-symptomatic stage. In turn, this probably indicates that the initiation of an early clinical evaluation and antiviral and/or mAbs treatment should be suggested.

In fact, as previously discussed, the major threat of COVID-19 is the “cytokine storms”, a hyperactivation of the immune system and the uncontrolled release of cytokines that can cause severe acute respiratory distress syndrome (ARDS) [41]. In this clinical phase, the use of antiviral therapy has limited effects, while only the use of potent anti-inflammatory drugs, including corticosteroids (dexamethasone), anti-cytokines, monoclonal antibodies (tocilizumab, anakinra), and small molecules (baricitinib), may be beneficial [1,2,3,4,5,6,7,37]. However, the precise timing of administration and the safety profile of these drugs are still under discussion, highlighting the importance of preferring different treatment strategies that could prevent the progression of the disease from the pre-symptomatic phase to a cytokine storm, such as antiviral drugs.

Furthermore, the importance of vaccination should not be forgotten: in an exploration of the predictors of COVID-19 disease progression, a complete vaccination course was found to be the strongest protective factor, in line with the current literature [42,43,44]; in our series, three doses also proved protective at multivariable analysis, while two doses did not improve the outcome. This could be related to the low protective effect of an incomplete vaccination schedule but could also be related to the low number of patients enrolled with only two doses.

Severe secondary bacterial infections were also confirmed as independent predictors for serious form of COVID-19. Notably, secondary infections are associated with an increase in the mortality risk in patients hospitalized for SARS-CoV-2 infections [45,46,47,48]; hence, the adoption of measures for infection prevention and control is still pivotal in COVID-19 hospitals and should be implemented where lacking [48,49].

This study has some limitations: firstly, the retrospective nature of the research. Secondly, this is a monocentric study: these results should be externally validated. Moreover, no patient in the study cohort was treated with oral antiviral drugs; however, the use of these medications in hospitalized patients should be further investigated. In this work, the use of a combination therapy of remdesivir and monoclonal antibodies was explored; similarly, combination therapies with two antiviral drugs should be evaluated and compared with other strategies. Finally, the variant of concern was unknown in some patients, and the choice of monoclonal antibodies was based on epidemiological data periodically published by the Italian Ministry of Health [50].

## 5. Conclusions

In conclusion, this work suggests that REM and monoclonal antibodies may reduce the risk of disease progression in patients hospitalized for mild–moderate COVID-19. In addition, combination therapies could be beneficial for seronegative subjects and particularly for severely immunocompromised patients: further studies should address the importance of tailoring the antiviral therapies according to the comorbidities of patients and their immunological status.

## Figures and Tables

**Figure 1 viruses-15-01199-f001:**
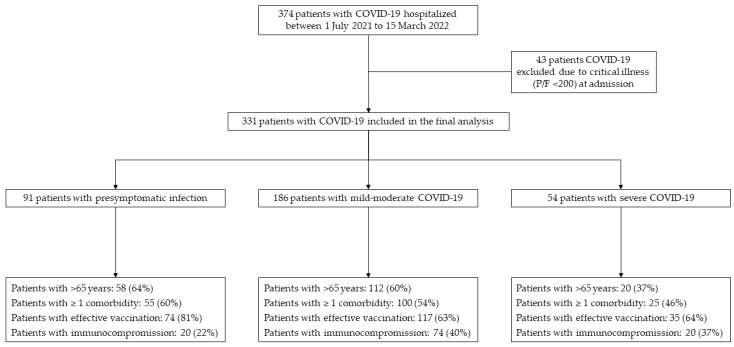
Retrospective enrollment flowchart.

**Figure 2 viruses-15-01199-f002:**
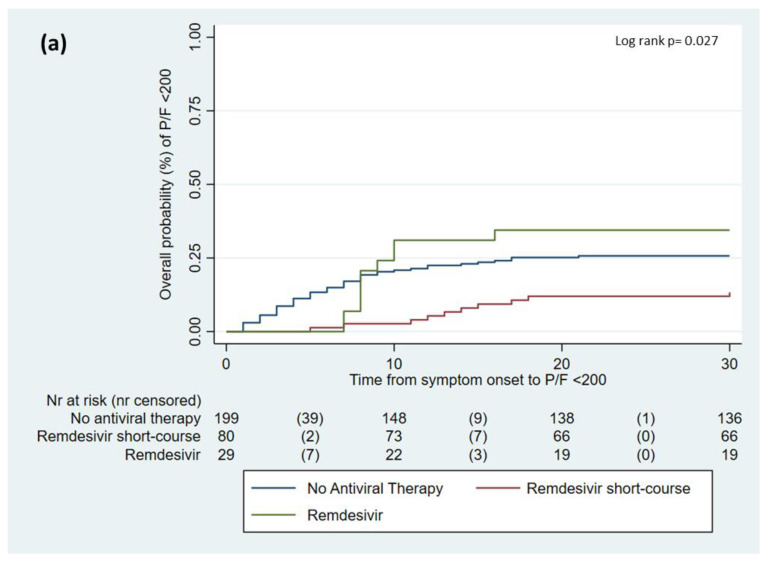
Impact of different remdesivir administration schedules (**a**) and vaccination (**b**) on risk of COVID-19 progression.

**Figure 3 viruses-15-01199-f003:**
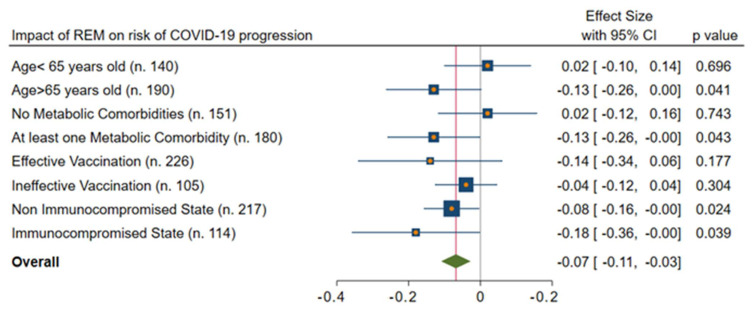
IPTW-adjusted analysis of impact of REM on risk of COVID-19 progression. Legend: REM: remdesivir.

**Figure 4 viruses-15-01199-f004:**
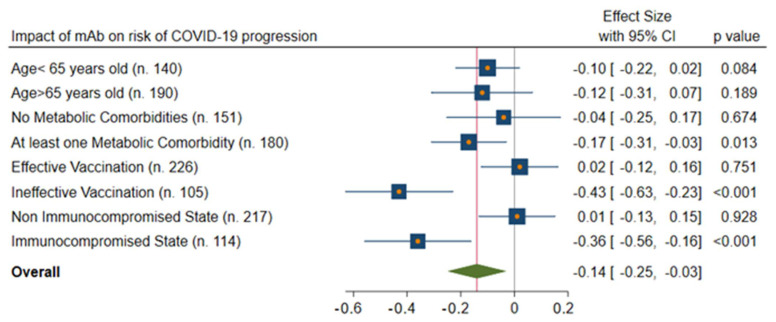
IPTW-adjusted analysis of impact of mAbs on risk of COVID-19 progression. Legend: mAbs: monoclonal antibodies.

**Figure 5 viruses-15-01199-f005:**
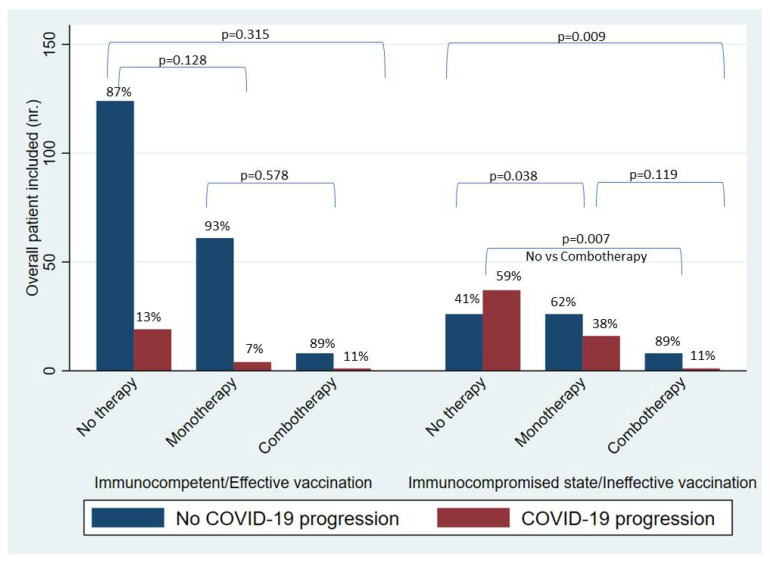
Impact of mono- and combo-therapy on risk of COVID-19 progression. Legend: Immunocompetent/Effective vaccination: No therapy (n. 143); monotherapy (n. 65); combo-therapy (n. 11). Immunocompromised (n. 114): No therapy (n. 63); monotherapy (n. 42); combo-therapy (n. 11).

**Table 1 viruses-15-01199-t001:** Preestablished internal COVID-19 treatment protocol.

Patients with Asymptomatic or Mild to Moderate SARS-CoV-2 Infection with no Need for Oxygen Therapy	Patients with Severe SARS-CoV-2 Infection and Respiratory Failure (P/F < 300)	Critical SARS-CoV-2 Infection Requiring High Flow Oxygen Therapy/Non-Invasive Ventilation
Antiviral Therapy	Antiviral Therapy	Antiviral Therapy
**Remdesivir 3-day short course**	**Remdesivir 5-day course**	**None**
**Inclusion criteria:**		
Solid tumor/ hematologic neoplasia		
Chronic kidney failure		
Diabetes		
BMI > 30		
Severe cardiovascular disease	**Inclusion criteria:**	
Chronic liver impairment	Radiological pneumonia signs	
Any condition of immunosuppression	Symptom onset <10 days (even >10 days in immunocompromised)	
**Monoclonal Antibodies**	**Monoclonal Antibodies**	**Monoclonal Antibodies**
**Inclusion criteria:**	**None**	**None**
Ineffective vaccination and severe kidney/liver impairment that contraindicated remdesivir		
**OTHER THERAPIES**	**OTHER THERAPIES**	**OTHER THERAPIES**
**Low molecular weight heparin** for 14 days or until discharge	**Low molecular weight heparin** for 14 days or until discharge	**Low molecular weight heparin** for 14 days or until discharge
	**PLUS**	**PLUS**
	**Dexamethasone** for 10 days	**Dexamethasone** for 10 days
**Immunocompromised state or multiple risk factors in subjects with ineffective vaccination**	**Immunocompromised state or multiple risk factors in subjects with ineffective vaccination**	**Immunocompromised state or multiple risk factors in subjects with ineffective vaccination**
**Remdesivir + mAbs**	**Remdesivir + mAbs**	** No Antiviral/mAbs **

Legend: BMI: body mass index; mAbs: monoclonal antibodies.

**Table 2 viruses-15-01199-t002:** General characteristics of the study population.

	Overall (n. 331)	No Disease Progression (n. 253)	Disease Progression (n. 78)	*p*-Value
**Age (y), median (q1–q3)**	71 (51–80)	66 (48–79)	75 (65–84)	**0.005**
**Male sex, n (%)**	173 (52)	138 (55)	35 (45)	0.135
**Metabolic comorbidities, n (%)**				
*Cardiovascular diseases*	32 (10)	22 (9)	10 (13)	0.281
*Diabetes*	70 (21)	49 (19)	21 (27)	0.153
*Chronic lung diseases* *	50 (15)	40 (16)	10 (13)	0.519
*Chronic kidney failure*	28 (8)	18 (7)	10 (13)	0.113
*Chronic liver diseases*	14 (4)	10 (4)	4 (5)	0.652
*Obesity*	48 (14)	36 (14)	12 (15)	0.800
**Immunosuppressive state, n (%)**				
*Autoimmune disease*	14 (4)	11 (4)	3 (4)	0.847
*Any solid cancer*	62 (19)	49 (19)	13 (17)	0.593
*Hematologic neoplasia*	23 (7)	18 (7)	6 (7)	0.768
*Solid organ transplant recipient*	15 (5)	10 (4)	5 (6)	0.316
**Vaccination status, n (%)**				
*No or one dose of anti-SARS-CoV2 vaccine* **	80 (24)	45 (18)	35 (45)	**<0.001**
*Two doses of anti-SARS-CoV2 vaccine*	25 (8)	8 (3)	17 (22)	
*Three doses of anti-SARS-CoV2 vaccine*	226 (68)	200 (79)	26 (33)	
**Median (q1–q3) months since last dose of vaccine**	3 (2–5)	3 (3–5)	2 (2–4)	0.145
**Symptoms at hospitalization, n (%)**				
*Asymptomatic*	91 (27)	82 (32)	9 (12)	**<0.001**
*Fever and organ symptoms*	113 (34)	93 (37)	20 (26)	0.070
*Pneumonia and hypoxia*	30 (9)	12 (5)	18 (23)	**<0.001**
*Other symptoms*	97 (30)	64 (26)	31 (39)	**0.020**
**COVID-19 disease severity on admission, n (%)**				
*Presymptomatic infection*	91 (27)	82 (32)	9 (12)	
*Mild–moderate illness*	186 (56)	128 (51)	58 (74)	**<0.001**
*Severe illness*	54 (17)	43 (17)	11 (14)	
**Laboratory features at admission, median (q1–q3)**				
*Leucocytes, cell/uL*	6310 (4680–9030)	6240 (4770–8810)	7270 (4540–9600)	0.357
*Lymphocytes, cell/uL*	1089 (677–1560)	1114 (695–1630)	916 (650–1218)	0.069
*Platelets, cell/uL*	196 (149–258)	197 (149–255)	196 (143–269)	0.783
*C-reactive protein mg/L*	62 (13.3–112)	53.9 (11.5–97.1)	131 (72.9–178)	**<0.001**
**Treatment included in standard of care, n (%)**				
*Corticosteroids*	185 (56)	119 (47)	66 (85)	**<0.001**
*Low molecular weight heparin/other anticoagulants*	244 (74)	175 (69)	69 (88)	**0.001**
**Treated with monoclonal antibody, n (%)**	37 (11)	33 (13)	4 (5)	0.052
**Treated with remdesivir, n (%)**				
*No antivirals*	211 (63)	155 (62)	56 (72)	**0.011**
*Short-course remdesivir*	88 (27)	77 (30)	11 (14)	
*remdesivir*	32 (10)	21 (8)	11 (14)	
**Combination therapy (REM plus mAbs), n (%)**	18 (5)	16 (6)	2 (3)	0.108
**Complications during hospitalization, n (%)**				
*Heart failure*	25 (8)	16 (6)	9 (12)	0.128
*Kidney failure*	22 (7)	16 (6)	6 (8)	0.678
*Secondary severe infection*	33 (10)	17 (7)	16 (21)	**<0.001**
**All-cause in-hospital mortality, n (%)**	47 (14)	19 (7)	28 (36)	**<0.001**

Legend: q1–q3: first–third quartile; REM: remdesivir; mAbs: monoclonal antibodies; *: including 44 chronic obstructive lung diseases, 5 severe asthma, and 1 idiopathic pulmonary fibrosis; **: only 5 patients with 1 vaccine dose.

**Table 3 viruses-15-01199-t003:** Predictors of COVID-19 progression (P/F < 200) (n. 331).

		Univariate Analysis			Multivariate Analysis	
	HR	95% CI	*p*-Value	aHR	95% CI	*p*-Value
**Age (per 1 year increase)**	1.02	1.00–1.03	**0.001**	1.04	1.01–1.06	**<0.001**
**Male sex**	0.79	0.43–1.27	0.334	0.71	0.37–1.37	0.319
**At least one comorbidity**	1.57	0.93–2.65	0.088	0.93	0.45–1.93	0.866
*Cardiovascular diseases*	1.15	0.72–1.86	0.543	\		
*Diabetes*	1.51	0.89–2.56	0.124	\		
*Chronic lung diseases*	0.69	0.33–1.44	0.325	\		
*Chronic kidney failure*	1.46	0.66–3.19	0.340	\		
*Obesity (BMI > 30)*	1.07	0.56–2.04	0.835	\		
*Any solid cancer*	0.75	0.39–1.43	0.390	\		
*Hematologic cancer*	1.20	0.52–2.77	0.666	3.62	1.00–13.05	**0.049**
**Vaccination status**						
*No or one dose of anti-SARS-CoV2 vaccine*	1			1		
*Two doses of anti-SARS-CoV2 vaccine*	2.73	1.05–7.05	**0.038**	2.26	0.70–7.24	0.169
*Three doses of anti-SARS-CoV2 vaccine*	0.16	0.09–0.30	**<0.001**	0.11	0.05–0.24	**<0.001**
**COVID-19 disease severity on admission**						
*Presymptomatic infection*	1			1		
*Mild–moderate illness*	4.12	1.94–8.78	**<0.001**	4.70	1.86–11.83	**0.001**
*Severe illness*	2.33	0.89–6.05	0.082	3.35	1.11–10.12	**0.032**
**Complications during hospitalization**						
*Heart failure*	1.63	0.74–3.58	0.216	\		
*Kidney failure*	1.22	0.49–3.05	0.658	\		
*Secondary severe infection*	3.31	1.86–5.87	**<0.001**	3.36	1.30–8.63	**0.012**
**Monoclonal antibody**	0.43	0.15–1.18	0.104	\		
**Antiviral therapy**						
*No antiviral*	1			1		
*Short-course remdesivir*	0.44	0.22–0.88	**0.021**	0.20	0.08–0.49	**<0.001**
*Remdesivir*	1.29	0.65–2.56	0.454	0.75	0.27–2.04	0.578
**Treatment strategy**						
*No therapy*	1			\		
*Monotherapy (antiviral or mAb)*	0.65	0.38–1.12	0.127	\		
*Combination therapy (antiviral plus mAb)*	0.42	0.10–1.74	0.236	\		

Legend: mAbs: monoclonal antibodies.

**Table 4 viruses-15-01199-t004:** Predictors of COVID-19 progression (P/F < 200) in immunocompromised patients or in patients with ineffective vaccination (n. 114 *).

		Univariate Analysis			Multivariate Analysis	
	HR	95% CI	*p*-Value	aHR	95% CI	*p*-Value
**Age (per 1 year increase)**	1.02	1.00–1.03	**0.006**	1.04	1.01–1.07	**0.006**
**Male sex**	0.69	0.39–1.23	0.213	0.37	0.14–1.00	0.050
**At least one comorbidity**	1.92	0.90–4.05	0.087	0.97	0.32–2.97	0.970
*Cardiovascular diseases*	1.23	0.68–2.20	0.479			
*Diabetes*	2.31	1.20–4.44	0.012			
*Chronic lung diseases*	1.32	0.41–4.27	0.634			
*Chronic kidney failure*	1.37	0.54–3.46	0.500			
*Obesity (BMI > 30)*	0.63	0.27–1.49	0.300			
*Any solid cancer*	0.67	0.31–1.43	0.307			
*Hematologic cancer*	0.66	0.26–1.67	0.387	1.29	0.27–6.11	0.747
**COVID-19 disease severity on admission**						
*Presymptomatic infection*	1			1		
*Mild–moderate illness*	5.88	1.78–19.28	**0.004**	13.09	2.82–60.77	**0.001**
*Severe illness*	1.71	0.40–7.33	0.468	5.93	0.89–39.54	0.066
**Complications during hospitalization**						
*Heart failure*	1.87	0.67–5.23	0.229			
*Kidney failure*	1.37	0.33–5.64	0.662			
*Secondary severe infection*	2.78	1.34–5.76	**0.006**	1.91	0.41–8.76	0.402
**Monoclonal antibody**	0.19	0.04–0.80	**0.024**	\		
**Antiviral therapy**						
*No antiviral*	1			\		
*Antiviral therapy with remdesivir*	0.51	0.28–0.91	**0.025**	\		
**Treatment strategy**						
*No therapy*	1			1		
*Monotherapy (antiviral or mAb)*	0.47	0.25–0.87	**0.018**	0.23	0.08–0.65	**0.006**
*Combination therapy (antiviral plus mAb)*	0.13	0.19–1.02	0.053	0.06	0.02–0.77	**0.002**

Legend: *: autoimmune diseases 14 (13%); any solid cancer: 62 (54%); hematologic neoplasia: 23 (20%); solid organ transplant recipient: 15 (13%); mAbs: monoclonal antibodies.

## Data Availability

The dataset used for analysis is available from the corresponding author on reasonable request.

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
