# Peer review of "Efficacy of Remdesivir and Neutralizing Monoclonal Antibodies in Monotherapy or Combination Therapy in Reducing the Risk of Disease Progression in Elderly or Immunocompromised Hosts Hospitalized for COVID-19: A Single Center Retrospective Study"

_viruses, 2023, doi:10.3390/v15051199_

Round 1

Reviewer 1 Report (Previous Reviewer 2)

The authors have revised the manuscript and the quality has improved. It focused on the old and immunodeficiency patients and combination therapy, which may provide some evidence of new therapy for this special population.

Author Response

Reviewer 1#:

The authors have revised the manuscript and the quality has improved. It focused on the old and immunodeficiency patients and combination therapy, which may provide some evidence of new therapy for this special population.

We really thank the reviewer for his kind words and the appreciation of our work. We hope that this work will improve clinical practice and inspire future works.

Reviewer 2 Report (New Reviewer)

The article "COVID-19 into chemical science perspective: From receptor binding of the virus to chemical preventive measures and drug development" attempts to highlighting the potential value of combination therapy in combating the pandemic. The paper is well structured. I recommend this article to be published in the journal. Here are some suggestions:

1. Keywords” should be improved.

2. In lines 56-58, “Initial research was focused on multiple drugs, including corticosteroids [1], immunomodulatory therapy [2], and monoclonal antibodies [3-5]”, literature survey can be improved by including the article appeared during the review process as well as few other relevant articles which author miss. For example, “Initial research was focused on multiple drugs, including corticosteroids [1], immunomodulatory therapy [2], Chinese herbal medicine (DOI: 10.1016/j.jep.2021.113869), natural products (DOI: 10.3390/v13020305), and monoclonal antibodies [3-5]”

3. In “Introduction”, As new variants continue to evolve, it is important to discuss the drug resistance of Remdesivir. In fact, Remdesivir drug resistance has been extensively studies. Please refer to: DOI: 10.1038/s41467-022-29104-y; DOI: 10.3389/fimmu.2022.1015355. Combination therapy could address drug resistance conferred by emerging variants.  

4. For the benefits of the readers please list other Combination strategy for SARS-CoV-2 treatment. For example, Natural Products in Combination with FDA-Approved Drugs to Treat COVID-19 (DOI: 10.3390/biomedicines9060689). Natural products have demonstrated potential value and with the assistance of nanotechnology, combination drug therapies, and the prodrug strategy, this “natural remedy” could serve as a starting point for further drug development in treating COVID-19.

5. In “Conclusions” section, the quality of the Conclusions must be improved.

Editing of English language required.

Author Response

Reviewer 2#:

The article "Efficacy of Remdesivir and Neutralizing Monoclonal Antibodies in Monotherapy or Combination Therapy in Reducing the Risk of Disease Progression in Elderly or Immunocompromised Hosts Hospitalized for COVID-19: a Single Center Retrospective Study" attempts to highlighting the potential value of combination therapy in combating the pandemic. The paper is well structured. I recommend this article to be published in the journal. Here are some suggestions:

We thank the reviewer for his kind words on our paper and his positive judgment on our work. We hope that the manuscript will be further improved after revisions.

  1. “Keywords” should be improved.

            As suggested, Keywords were improved as follows:

“SARS-CoV-2; Remdesivir; Sotrovimab; Antivirals; Monoclonal Antibodies; Severe COVID-19; Immunocompromised Hosts; Combination Therapy; Elderly; COVID-19 progression; COVID-19 therapy.”

  1. In lines 56-58, “Initial research was focused on multiple drugs, including corticosteroids [1], immunomodulatory therapy [2], and monoclonal antibodies [3-5]”, literature survey can be improved by including the article appeared during the review process as well as few other relevant articles which author miss. For example, “Initial research was focused on multiple drugs, including corticosteroids [1], immunomodulatory therapy [2], Chinese herbal medicine (DOI: 10.1016/j.jep.2021.113869), natural products (DOI: 10.3390/v13020305), and monoclonal antibodies [3-5]”

            The sentence was rephrased as suggested and the two citations were included.

  1. In “Introduction”, As new variants continue to evolve, it is important to discuss the drug resistance of Remdesivir. In fact, Remdesivir drug resistance has been extensively studies. Please refer to: DOI: 10.1038/s41467-022-29104-y; DOI: 10.3389/fimmu.2022.1015355. Combination therapy could address drug resistance conferred by emerging variants.

            As suggested, we implemented the following sentence along with the two citations indicated:

“Finally, also the progressive the novo or iv vivo emergence of viral variants showing resistance to antiviral therapies needs to be addressed in future [DOI: 10.1038/s41467-022-29104-y], along with the development of new drugs for combating COVID-19 [DOI: 10.3389/fimmu.2022.1015355]. In this sense, it could be speculated that combination therapy could address drug resistance conferred by emerging variants, but data are still very preliminary [DOI: 10.3390/microorganisms10071475]”

  1. For the benefits of the readers please list other Combination strategy for SARS-CoV-2 treatment. For example, Natural Products in Combination with FDA-Approved Drugs to Treat COVID-19 (DOI: 10.3390/biomedicines9060689). Natural products have demonstrated potential value and with the assistance of nanotechnology, combination drug therapies, and the prodrug strategy, this “natural remedy” could serve as a starting point for further drug development in treating COVID-19.

            As suggested, we included the following sentence:

“Future studies should also address the potential role of other molecules in combination with current therapies, for instance those deriving from nanotechnologies [DOI: 10.3390/biomedicines9060689]”

  1. In “Conclusions” section, the quality of the Conclusions must be improved.

            Conclusions were rephrased as follows:

“In conclusion, this work suggests that REM and monoclonal antibodies may reduce risk of disease progression in patients hospitalized for mild-moderate COVID-19. In addition, combination therapy could be beneficial for seronegative subjects and particularly for severely immunocompromised patients: further studies should address the importance of tailoring the antiviral therapies according to comorbidities of patients and their and immunological status.”

Editing of English language required.

            The paper was revised by a native English speaker and several grammar mistakes were corrected.

This manuscript is a resubmission of an earlier submission. The following is a list of the peer review reports and author responses from that submission.

Round 1

Reviewer 1 Report

Davide et al, showed a report regarding the efficacy of Remdesivir and Monoclonal antibody therapy for the patients with COVID-19 including immunocompromised subjects in the real-life settings.

This manuscript is well-written and revealed the situation of treatment of COVID-19 in Italy. However, I think some points of the material covered in the current paper has been addressed in prior publication, additionally, new insight of treatment strategies should be discussed. Therefore, I would like to suggest several points should be modified.

Major comment

1.      In Method section, Authors defined the “ineffective vaccination” as the patients who received COVID-19 vaccination less than 3 times. However, for the patients with delta variant, two doses of vaccination is somehow effective and the timing of the infection from last vaccination is uncertain. I recommend authors should identify times of vaccination (not less than 3times, but 0,1,2 times)

2.      The authors defined patients with solid cancer as immunocompromised status. All of them were in “active” cytotoxic chemotherapy? Active chemotherapy should be clarified how many days or months after the last chemotherapy.

Minor comments

1.      Page 3, line 121. The word “ECMO” is the first time described, full name should be noted. (extra-corporeal membrane oxygenation).

2.      Page 6, line 223. Afterall, what kind of adverse event happened? The author did not describe about adverse events. This is the secondary outcome, so it should be discussed more.

3.      Many conjunctions were used at the beginning of the sentences everywhere.it seems overuse. The authors should use conjunctions as appropriate.

Reviewer 2 Report

For people infected with COVID-19 with high risk of disease progression, the use of remdesivir or monoclonal antibody is one of the standard treatment measures recommended by the guidelines. Recently, WHO and other guidelines recommended that remdesivir can also be used for patients with severe COVID-19, but the current guidelines do not recommend the use of monoclonal antibody. In addition, the baseline disease severity of the two groups of patients in this study was different, and the risk of disease progression itself is different. In addition, the use of remdesivir or monoclonal antibody by these patients was also based on the guidelines at that time, and there was a serious imbalance in the baseline distribution of the two groups of patients. The authors did not include the variable of the severity of the baseline condition in the univariate analysis and multivariate analysis. The total number of cases in the classification of symptoms at admission in Table 1 is 333, while 331 patients were enrolled in this study, with obvious errors. 

Reviewer 3 Report

The article is well thought out and explains the objectives of the study but I see the irrelevance of this therapy with currently available therapies and the only reason for publishing it is the old age population studied in this study, so I highly recommend changing the title of the study, secondly add relevant literature with reference to age (old-age) and strengthen your article from that side too. Immunocompromised is another good factor included in this study, actually, I really liked the inclusion and exclusion criteria of the study, so I highly recommend highlighting the novelty of the study. I would also request to add Chronic obstructive pulmonary disease (COPD) in comorbidities if you can. Cytochrome stormes was a major discussion point, which is some bit ignored.

minors: Statistical analysis is excellent but English much be improved. abstract is not properly written and is hard to understand. line 69 has double full stops.